# The Utility of Intraluminal Therapies in Upper Tract Urothelial Carcinoma: A Narrative Review

**DOI:** 10.3390/cancers16101931

**Published:** 2024-05-18

**Authors:** Jack Tyrrell, William Chui, Joshua Kealey, Shomik Sengupta

**Affiliations:** 1Urology Department, Eastern Health, Box Hill, Victoria 3128, Australia; 2Eastern Health Clinical School, Monash University, Box Hill, Victoria 3128, Australia

**Keywords:** upper tract urothelial carcinoma, transitional cell carcinoma, intraluminal therapy, stent, nephrostomy

## Abstract

**Simple Summary:**

Tumours of the lining of the kidney and its drainage tube (the ureter) can be treated by surgery carried out through fine telescopes. There is an increased risk of tumours growing back after such surgery. One of the ways to reduce the risk is to put medications such as chemotherapy into the kidney to treat the lining and tumours on it. The location and structure of the kidney make this a difficult process. This article outlines the available and developing options for such treatment.

**Abstract:**

Nephron sparing surgery (NSS) is considered for selected cases of upper tract urothelial carcinoma (UTUC) as it maintains renal function and avoids morbidity associated with radical nephroureterectomy (RNU). The appropriate selection of patients suitable for NSS without compromising oncological outcomes can sometimes be difficult, given the limitations of diagnostic modalities. Recurrence rates for UTUC can be as high as 36 to 54% after NSS. Intraluminal adjuvant therapy can be attempted following NSS to reduce recurrence, but delivery to the upper tract is more challenging than into the bladder. Bacillus Calmette-Guerin (BCG) and chemotherapy such as Mitomycin (MMC) have been administered via nephrostomy or ureteric catheter, which requires invasive/repeated instrumentation of the upper urinary tract. Drug delivery by reflux from bladder instillation along indwelling stents has also been tried but can potentially be unreliable. Recently, a gel formulation of mitomycin has been developed for the controlled exposure of the upper urinary tract to treatment over a number of hours. Drug-eluting stents to deliver chemotherapy to the upper urinary tract have been developed but have not yet entered clinical practice. Endoluminal phototherapy utilising an intravenous photosensitising agent is another novel approach that has recently been described. Intraluminal therapies may be beneficial in decreasing recurrence rates in UTUC, but currently have some limitations in their usage.

## 1. Introduction

Upper Tract Urothelial Carcinoma (UTUC) is a relatively rare malignancy of urothelial origin arising between the ureteric orifice and the renal collecting system. UTUC accounts for only 5–10% of all urothelial cancers and 10% of renal tumours [1]. The estimated annual incidence is two cases per 100,000 people in Western countries. Patients with UTUC are diagnosed at an older age (with a mean age of diagnosis of 73.4 years) and with more invasive disease than those with bladder cancer [2]. Although there are similarities between UTUC and bladder-based urothelial cancers, diagnosis, management and outcomes differ greatly.

Radical nephroureterectomy remains the gold-standard treatment for non-metastatic high-risk UTUC. However, for localised low-grade disease, kidney sparing surgery (segmental ureterectomy or endoscopic management by retrograde or percutaneous antegrade approach) can be utilised, as it minimises morbidity associated with RNU without compromising the oncological and survival outcomes. In order to be considered low-risk, tumours must be unifocal, <2 cm, have negative cytology, have a low-grade histology and appear non-invasive on computed tomography CT [3]. In recent years, the use of intraluminal therapies has been investigated as an adjuvant to kidney sparing surgery with the aim of reducing recurrence rates. The therapies can be delivered directly into the urinary tract in either an antegrade or a retrograde manner. This review article will outline the current landscape and potential future directions for intraluminal therapies in UTUC.

## 2. Need for Intraluminal Therapy

While radical nephroureterectomy remains the gold standard for the treatment of UTUC, it is an invasive procedure with considerable morbidity. Reported complication rates range from 12.2–18.6%, with bleeding requiring blood transfusion, renal failure, surgical site infection, sepsis and mortality as significant risks. The risk of a Clavien Dindo grade IV complication and 30-day mortality rates are 5.3% and 1.7%, respectively [4].

Kidney sparing surgery (KSS) may be considered in appropriately selected patients to minimise the morbidity associated with RNU. It is best utilised in those with low-risk disease and those in whom concomitant medical problems preclude them from RNU (Table 1). These include those with a single kidney, bilateral disease, severe chronic kidney disease, and significant medical comorbidities, making them unsuitable for major surgery.

Unfortunately, recurrence rates for UTUC remain high after RNU and KSS. One systematic review analysing KSS reported a rate of bladder recurrence at 17–35% and upper tract recurrence at 36–54% [5]. Lindner et al. found that the overall recurrence rate in the KSS group was 57%. While all these patients had low-grade biopsies pre-operatively, only 78% (11/14) were low-grade on the final histology, indicating a significant rate of upgrading [6]. The high recurrence rates post KSS highlight the need for carefully selected application as well as effective and safe intraluminal adjuvant therapies that may reduce recurrence and progression.

## 3. Difficulties in Optimal Patient Selection

Selecting which patients should undergo RNU versus KSS can be difficult. There are several investigations that are part of the decision-making process. Current EAU guidelines recommend computed tomography (CT) alongside urine cytology to help diagnose UTUC. Cystoscopy is also used to rule out concurrent bladder cancer. Diagnostic ureteroscopy is usually used for confirmation, especially if the diagnosis is unclear, and to help stratify risk by histological stage and grade [3].

CT urography has a higher sensitivity (92%) and specificity (95%) when compared to magnetic resonance imaging (MRI) for diagnosing UTUC, as well as being easier to access and less time-consuming in most centres. However, flat lesions such as carcinoma in situ can be missed by both modalities, as they may not show any urothelial thickening or a space-occupying lesion on imaging. The level of invasion of tumours is hard to distinguish, with only 67% of infiltrating tumours being correctly diagnosed on imaging alone, likely due to microinvasion, which is not visible on CT [7].

Diagnostic ureteroscopy (dURS) can help facilitate the visualisation and histological assessment of tumours. One study found the sensitivity and specificity to be 84% and 83% when urologists were shown visual findings of dURS and asked to distinguish between tumour presence, no tumour or suspected CIS. However, 16% (16/97) of lesions were missed, with 93.5% (15/16) of these being CIS [8]. Endoscopic biopsies may allow histological confirmation of malignancy as well as help determine grade and stage. Unfortunately, these samples can often be non-diagnostic or difficult to definitively stage or grade, given the small sample size and issues in processing.

One review found the rate of insufficient biopsy tissue can vary between 4 and 37% [9]. Tumours can often be under-graded as well, meaning those initially receiving KSS for presumed low-grade UTUC were subsequently found to have high-grade cancer on the tumour specimen, leading to a requirement for repeat procedures and delays in definitive treatment. In addition, 44% of tumours were upgraded when their initial dURS biopsy was compared to their final histology [9]. Carcinoma in situ can often be missed on biopsy, whether in suspicious areas or random in distribution. Grahn et al found that dURS biopsies failed to diagnose UTCIS in up to 84.7% (29/39) of patients in a retrospective analysis [8].

Novel techniques have been proposed in an attempt to increase the quality and, thus, the accuracy of biopsy samples taken during dURS. Cryobiopsy involves creating an ice ball around the desired biopsy site with a cryoprobe. This ice ball is then more adherent to the probe in comparison to the surrounding tissue. One ex vivo study has shown that the use of cryobiopsy produces larger sized biopsy samples with less artefacts [10]. However, stronger evidence from in vivo studies in the clinical setting is still lacking.

Various new endoscopic enhancement techniques have also shown promise in increasing the accuracy of UTUC diagnosis and grading. Photodynamic diagnosis and narrow band imaging are often used in the diagnosis of bladder cancers and are similarly being introduced to upper tract lesions. Confocal laser endomicroscopy is a novel fluorescence-based fibre optic technique, and in a small study it was able to accurately differentiate between low-grade (100%), high-grade (83%) and CIS (100%) tumours [9].

A systematic review by Marchioni et al. also reported a significant increase in the rate of intravesical recurrence in those who underwent dURS and biopsy prior to RNU, being 39.2–60.7% compared to 16.7–46% in those who did not [11]. Hence, in an attempt to reduce the necessity of dURS, various non-invasive tests have been investigated to aid in the diagnosis of UTUC. One systematic review by Bialek et al. looked at a range of non-invasive urinary and blood biomarkers [12]. They reported that although voided cytology had a good specificity (54–100%), it was not overly sensitive (11–71.1%). They also found that fluorescence in situ hybridisation had a much greater sensitivity (73.7–87.5%) while maintaining good specificity (80–89.8%).

Other urinary tests were investigated in smaller studies, including thos detecting various proteins in urine samples, such as Bladder Tumour Antigen (BTA) protein, Nuclear Matrix Protein 22 (NMP22) and Immunocyt/uCyt+. Although initially designed for bladder cancer, Territo et al. investigated the use of EpiCheck in UTUC, a urine test comprised of 15 DNA methylation biomarkers, with results of 83% sensitivity and 79% specificity [13]. MicroRNAs as serum biomarkers have also been examined, although due to heterogeneity in groups selected, cancer stage and histological grade, diagnostic utility varied greatly in sensitivity (29.5–97.8%) and specificity (29.4–100%) [12].

## 4. Agents in Use for Intraluminal Therapy

Intraluminal therapies for UTUC can involve the same chemotherapeutic and immunotherapy agents that are often used in the management of urothelial tumours in the bladder. These can include Bacille Calmette-Guerin (BCG), Mitomycin (MMC), Epirubicin, Thiotepa and Gemcitabine, among others. The more recent studies have generally focused on BCG or mitomycin use.

Mitomycin is often administered following transurethral resection of bladder tumours (TURBT) and RNU and is delivered directly into the bladder via a urinary catheter. Van Wijngaarden et al. found that Mitomycin scheduled over the course of a 6-month regimen significantly reduces the rate of bladder recurrence following RNU. They found recurrence-free survival rates at 1, 2 and 5 years to be 95%, 86% and 86% in those who received Mitomycin, in comparison to 67%, 63% and 43% in those who did not [14].

Extrapolating from this, intraluminal chemotherapy has also been applied after endoscopic procedures to reduce recurrences of UTUC. Gallioli et al. looked at the efficacy of a single dose of mitomycin delivered via either single-J or double-J stents following endoscopic ablation for UTUCs. They compared 17 patients who all received a single dose of MMC within 6 h of endoscopic resection to 18 patients who received no treatment. The MMC group had a near 50% relative reduction in recurrence within 3 years (23% vs. 56% for bladder recurrence and 17% vs. 33% for upper tract recurrence) as well as prolonging the recurrence free survival (28 vs. 18 months). Unfortunately, there was a 40% complication rate with those who received MMC, although it is unclear if these complications were associated with the procedure itself or directly caused by the MMC treatment [15].

Metcalfe et al. investigated induction and maintenance MMC for UTUC; however, due to heterogeneity between different patient maintenance regimens, it was difficult to reach any definite conclusions regarding the efficacy of maintenance M. Overall, 60% of patients had recurrence-free survival by the 3-year follow-up [16]. The promising results from the use of MMC to treat UTUC have led to the development of a thermal gel containing MMC [17].

Bacillus Calmette Guerin (BCG) is an immunotherapeutic agent commonly used in patients with high-grade, non-muscle-invasive bladder cancer and has similarly been used for non-invasive UTUC. Giannarini et al. carried out a retrospective analysis of patients who received BCG for UTUC, post-resection of Ta or T1 tumours and those with known upper tract carcinoma in situ (UTCIS). Recurrence rates were found to be lower in patients treated with curative intent for UTCIS (40% recurrence) when compared to patients treated for adjuvant treatment following endoscopic treatment of Ta and T1 tumours (59% recurrence) [18].

Redrow et al. carried out a systematic review looking into the efficacy and suitability of BCG for UTCIS and reported response rates of 64–100% [19]. However, the incorporated studies included in the analysis had different methods of instillation and different criteria for diagnosing UTCIS. Without standardised criteria for diagnosing UTIC, large multi-organisational studies are more difficult to implement, with some centres relying on positive cytology but negative endoscopic findings or biopsies, and some centres needing confirmatory biopsies for UTCIS. This study also reported no significant difference in disease-specific mortality or all-cause mortality when comparing patients with UTCIS who underwent RNU or received BCG treatment. 

Shapiro et al. treated biopsy-proven UTCIS using intrarenally instilled BCG and Interferon α2B through an open-ended ureteral catheter [20]. Although it was a small study with 11 patients, the results were encouraging. Eight patients achieved a complete response after six weekly treatments carried out in an office-based setting, with another two patients needing a second induction treatment to achieve a complete response (total of 10/11 91%). A total of nine (90%) patients had maintained a complete response by the time of their last follow-up, with only one patient having a recurrence of disease. No treatment-related adverse effects were noted.

## 5. Delivery Techniques and Limitations

There have been multiple delivery methods utilised for adjuvant intraluminal therapy; however, no conclusive evidence exists to suggest the superiority of any in particular. Instillation techniques include direct delivery antegrade via nephrostomy insertion; retrograde delivery via open-ended ureteric catheters or single J catheters; and retrograde delivery using double J stents to induce vesicoureteral reflux with intravesical instillation (Table 2).

The first use of the antegrade approach via nephrostomy tube in intraluminal therapies was by Studer and colleagues in 1989, when they used this technique to administer BCG for upper tract CIS [21]. The benefits of this approach include a reduction in bladder irritation when compared to other modes of administration and the ability to easily predict the distribution of the administered agent [22]. However, the potential drawbacks of this method include the risk of seeding tumours along the nephrostomy tract, drug leakage at the nephrostomy site, bacterial colonisation and sepsis from an open system and a poorer quality of life [23]. 

Conversely, the retrograde delivery approach with a ureteric catheter/single J stent was introduced by Sharpe and colleagues in 1993 [24]. Their rationale for choosing this method was to minimise retroperitoneal extravasation and improve contact time when compared to the antegrade nephrostomy method. They did identify that a drawback of this approach was the need for multiple visits for repeat cystoscopy and insertion of ureteric catheter for each cycle of treatment. 

Korkes et al. investigated the feasibility of using double J stents to deliver instilled fluids from the bladder to the upper urinary tract after inducing vesicoureteral reflux (VUR). They studied 51 patients who had stents inserted for stone disease, presumably a younger and fitter patient cohort than those with a diagnosis of UTUC. They found that those with a stent inserted for a longer period (average 21.6 days vs. immediately post-insertion) had a higher incidence of VUR, 87.5% vs. 51.4%. VUR was also correlated with volumes instilled in the bladder, with 63% having VUR induced with a bladder volume of 360 mL, whereas at 120 mL, VUR only occurred in 14% [22].

Having a double J stent inserted for a longer period may not be ideal due to side effects such as dysuria, intermittent haematuria and flank pain, which can be extremely bothersome in most patients, with some studies reporting up to a 80% incidence of stent-related symptoms [25]. The other issue arising from this study is the average volume typically instilled intravesically for therapy, which is 60 mL for BCG and 20 to 40 mL for Mitomycin, did not show any evidence of VUR. It is unclear if patients would be able to tolerate the bladder filled with up to 360 mL of BCG or Mitomycin solution for the 1–2 h treatment time as suggested by this study, and what the resulting consequences would be for dosing and toxicity. 

In Gallioli et al.’s study looking at single-dose Mitomycin, it was found that the 66% (2/3) patients who received Mitomycin via reflux along a double-J stent had a local recurrence in comparison to 7.7% (1/13) of those who received Mitomycin by direct delivery via a single-J catheter [15]. In Metcalfe et al.’s study, the rate of recurrence and progression of disease was higher in the patients who received their Mitomycin therapy via ureteral catheter (recurrences in 9/19 patients and progression in 4/19) in comparison to Mitomycinvia nephrostomy (recurrences in 2/9 and progression in 1/9), even though more low-grade tumours were treated with ureteral catheter (71%) compared to nephrostomy (29%) [16].

In addition, in vivo porcine models have been used to look at which technique is most efficacious in delivering intraluminal therapy to the upper tracts. Liu and colleagues used indigo carmine dye as a surrogate for intraluminal therapy and injected it into porcine kidneys [26]. Staining of the kidney was greatest via retrograde ureteric catheter, and this was statistically significant. Further studies will need to be conducted in humans before we can definitively conclude which modality is most efficacious in terms of drug delivery. 

## 6. New Developments and Future Directions

**Photodynamic therapy:** In addition to the established modalities of electro-cautery and laser, used widely for the endoscopic treatment of UTUC, novel techniques such as photodynamic therapy have recently been developed. Currently, there is a phase 3 multicentre trial (ENLIGHTED) recruiting patients to assess the efficacy of infrared light exposure for treating UTUC [27]. This was initiated based of the promising results on the phase 1 trial looking at the photosensitising agent Padeliporfin and vascular targeted photodynamic therapy [28]. 

Patients in the trial were administered a 10 min intravenous infusion of padeliporfin prior to having an infrared light delivered through a ureteroscope or pyeloscope for 10 min. A ureteral stent was inserted in select patients post procedure, which was removed later. Eighteen patients had one round of therapy prior to assessment; 9/18 (50%) achieved a complete response after only one treatment, and 8/18 (44%) had a partial response, meaning that the initial tumour was treated but there was evidence of other sites of disease in the ipsilateral kidney or ureter. Patients who partially responded were offered a second round of treatment, to which 4/8 had a complete response, giving an overall complete response in 13/18 (68%) patients 30 days after treatment. 

There were no ureteral strictures or stenosis noted post-treatment in these patients. Common adverse effects of treatment included flank pain, dysuria and haematuria, although ureteral stenting was believed to be a contributor to many of these. Both patients with high- and low-grade UTUC were included in the study, with low-grade cancers having a better complete response rate after 1 treatment (54% vs. 40%). The small sample size and limited follow up mean we cannot yet draw any conclusions about long-term durability or recurrence rates from this study, but it shows promising results early on.

**Mitomycin Gel:** One of the main challenges facing intraluminal therapy delivery is how to maximise drug concentration and contact time at the tumour site. Unfortunately, both of these factors are reduced by normal urine flow. To minimise this effect, a reverse thermal gel made with 4 mg/mL Mitomycin has been produced (Trade name: JELMYTO). This was approved by the US Food and Drug Administration on 15 April 2020 [17]. This thermal gel is a liquid at low temperatures, allowing ready instillation through a catheter or a nephrostomy tube. At body temperatures, it partially solidifies into a gel form that subsequently dissolves over the course of 4–6 h, thus providing longer contact time and more reliable delivery to the tumour site. 

The OLYMPUS trial [29] investigated patients with biopsy proven low-grade UTUC who received Mitomycin gel via a retrograde catheter into the renal pelvis. Fifty-nine percent (42/71) of patients achieved a complete response following the 6-week induction period, and 56% (23/41) of these maintained a complete response at 12 months. These patients received varying numbers of monthly maintenance cycles of treatment, with 17/23 with a durable response receiving ≧1 maintenance treatment and the remaining six receiving no maintenance therapy. 

However, treatment-related side effects were common and included ureteric stricture, flank pain, haematuria and urinary tract infections. Strictures were the most common and concerning adverse event, occurring in 31/71 (44%) of patients. This appeared to be related to the number of treatments, as 19/29 (66%) patients who received at least one maintenance treatment suffered ureteric stricture compared to 12/42 (29%) who did not receive any maintenance treatment. 

Rosen et al. investigated antegrade instillation of Mitomycin gel through a nephrostomy to help minimise the risk of patients suffering from ureteral strictures postprocedurally. Half of the patients (4/8) achieved a complete response after induction treatment with six weekly instillations. The other 50% had a partial response and went on to have their tumours endoscopically resected, including a patient whose tumour was previously unresectable due to its size [30].

Rose et al. administered Mitomycin gel via percutaneous nephrostomy to 32 patients over the course of 2 years, with only 9% (3/32) suffering from a stricture following induction therapy, compared to the 29% (12/41) stricture rate in the OLYMPUS study in patients who only received induction therapy. The low rates in Rose et al.’s study suggest that the strictures in the OLYMPUS trial may have been due to repeated ureteric procedures rather than the drug itself [31].

In these small-scale studies, the Mitomycin gel appears to be safe when delivered in an antegrade fashion for induction therapy.

**Drug-eluting stents**: Other methods of prolonging contact include drug-eluting or coated stents (DESs or DCSs). Such devices have been used in cardiology following percutaneous coronary intervention for ischaemic heart disease to help minimise the subsequent risk of re-stenosis. A recent systematic review assessed ureteric stents that similarly deliver antimicrobials, anti-inflammatory and anti-proliferative drugs. Preclinical in vitro and in vivo studies have shown promising results in terms of reducing inflammation, stricture rates and pain, as shown by the decreased use of anti-inflammatories for analgesia [32]. However, there have been limited human studies to date. 

Maintenance of drug concentrations at a high enough level to be efficacious remains a challenge. To help combat this, Soria et al. developed a stent coated in silk fibrin loaded with Mitomycin. This is a biodegradable stent (Braidstent^®^) with the aim of releasing Mitomycin continuously in a controlled manner as it degrades, with the further advantage of eliminating the need for cystoscopic stent removal. They found that it released Mitomycin over the course of 6 to 12 h, while the stent fully degraded within 14 weeks [33]. In a later study, Soria et al. proceeded to carry out the first live animal assessment of the silk fibrin mitomycin stent with pigs. They found that mitomycin was released for the first 12 h before the stent fully degraded by 7 weeks. However, ureteric stricture rates were 21%, and obstructive stent fragments were found in 28% of animals in the study [34].

Although a chemotherapy-eluting ureteric stent is not yet available for clinical use, a novel delivery device for the intravesical delivery of drugs has recently been developed. TAR-200 is a small silicone tube filled with Gemcitabine minitablets, which provides a sustained release of a low dose of the drug into the bladder over a period of 3 weeks. It can be inserted into the bladder using a proprietary introducer and removed using a flexible cystoscope [35].

A recently published study reported on the safety and preliminary efficacy of TAR-200 in patients with muscle-invasive bladder cancer (MIBC) [36]. Patients were given 4 cycles of TAR 200 used in 21-day cycles within a week of maximal transurethral resection of bladder tumour (TURBT), followed by three single quarterly cycles. Results were promising, with 31.4% (11/35) achieving a complete response and a further 8.6% (3/35) achieving a partial response following induction treatment. Progression-free survival in these patients was 92.3% at 6 months and 70.5% at 12 months. Side effects were reported in 40% of patients, most commonly dysuria, urinary frequency and nocturia. Overall, these encouraging results have led to further studies comparing the efficacy of TAR-200 (used with the PD-1 Inhibitor Cetrelimab) against BCG [37], chemoradiation therapy [38] and Cetrelimab alone [39]. 

A subsequent version of the device, the TAR-210, has now been developed to deliver Erdafitinib, a selective pan-FGR tyroskin kinase inhibitor that targets the FGFR3 mutation, which has been identified as an oncogenic driver in urothelial cancer [40]. An oral formulation of the drug has been approved for use in patients with locally advanced or metastatic urothelial cancers. The TAR-210 intravesical drug delivery system is now in phase 1 clinical trials with promising results. In a small cohort of intermediate- and high-risk patients with non-invasive bladder cancer (n = 43), 82% and 87% of patients achieved complete response, respectively [41]. Due to its molecular action, there is potential benefit from Erdafitinib administered intraluminally in the management of UTUC. These devices hold the promise that similar delivery devices configured appropriately for ureteric placement may someday enable the reliable delivery of intraluminal therapy for UTUC. 

## 7. Conclusions

Intraluminal therapies may be beneficial in decreasing the recurrence rates in patients with upper tract urothelial carcinoma following endoscopic kidney-sparing treatment. The potential benefits need to be weighed against difficulties in delivering intraluminal agents to the upper urinary tract, as well as a range of the side effects associated with the agents and the delivery techniques. Novel innovations, such as drug-eluting stents, are being investigated in an attempt to overcome some of these limitations.

## Figures and Tables

**Table 1 cancers-16-01931-t001:** Factors favouring nephron sparing surgery (NSS) for UTUC.

Low-risk disease: solitary, small, low-grade, non-invasive tumours
Single kidney
Bilateral disease
Severe chronic kidney disease
Significant medical co-morbidities

**Table 2 cancers-16-01931-t002:** Techniques for delivering intraluminal therapies.

Mode of Delivery		Advantages	Disadvantages
Antegrade	Nephrostomy	Reduction in bladder irritation symptoms Easier and more predictable distribution of medication	Tumour seeding along the nephrostomy tract Drug leakage alongside the nephrostomy Bacterial colonisation or sepsis in an open system Discomfort impacting the quality of life
Retrograde	Ureteric catheter or single-J stent	In vivo porcine model studies suggest superior calyceal coverage No quality-of-life impact from indwelling tubes	Repeated cystoscopy and ureteric catheter insertion
	Double-J stent	Administration directly into bladder via urethral catheter	Stent-related symptoms Unpredictable vesicoureteral reflux Requires larger volumes of the agent to be instilled

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
