# Peer review of "The Utility of Intraluminal Therapies in Upper Tract Urothelial Carcinoma: A Narrative Review"

_cancers, 2024, doi:10.3390/cancers16101931_

Round 1

Reviewer 1 Report

Comments and Suggestions for Authors

I read the article titled “ The Utility of Intraluminal Therapies in Upper Tract Urothelial 2 Carcinoma: A Narrative Review”.

The article is well written and provides an interesting overview on the topic.

I have two major comments:

-       I would improve the quality of the paper with a table for a more intuitive data reading.

-       While reading paragraph “3. Difficulties in Optimal Patient Selection” I notice that no data on urinary biomarkers in UTUC diagnosis/follow up were reported. I suggest the author to improve the discussion with relevant articles such as the following:

Territo A, Gallioli A, Diana P, Boissier R, Fontana M, Gaya JM, Sanguedolce F, Calderón J, Piana A, Fontanet S, Algaba F, Palou J, Breda A. DNA Methylation Urine Biomarkers Test in the Diagnosis of Upper Tract Urothelial Carcinoma: Results from a Single-Center Prospective Clinical Trial. J Urol. 2022 Sep;208(3):570-579. doi: 10.1097/JU.0000000000002748. Epub 2022 May 12. PMID: 35549312.

Author Response

Thank you for the comments:

  1. We have added 2 tables as per your suggestion
  2. We have added a section on the use of biomarkers, including suggested reference 

Reviewer 2 Report

Comments and Suggestions for Authors

The authors review current and possible future intraluminal treatment for urothelial carcinoma of the renal pelvis and ureter.  The review is comprehensive when covering the approaches of delivering intraluminal therapy. 

The authors should consider adding the following discussion points.

1. There are several references that report the limitations of ureteroscopic biopsy. It would interesting for the authors to discuss possible methods or technologies to improve the accuracy of the ureteroscopic biopses.

2. Please discuss why TAR 210 implants for upper tract tumors may be especially appropriate based upon mechanism of action and known tumor mutations

Author Response

Thank you for your kind comments 

We have addressed your suggestions:

  1. Section added on newer techniques to improve ureteroscopic diagnosis
  2. Detail added re underlying molecular basis for action of ertafinitib 

Reviewer 3 Report

Comments and Suggestions for Authors

I found this review article on intraluminal therapy for UTUC very interesting. Although some parts may seem a little wordy, it covers the latest information on clinical trials and contains content that will interest readers. Furthermore, the paragraph structure also aids the reader's understanding. This is a valuable context in which the difficulties of patient selection, drug selection, and the delivery system are explained. The conclusion feels a little redundant and could be summarized more clearly. I am concerned that readers may perceive the conclusion as emphasizing the usefulness of drug-eluting stents. It would be better to revise the Conclusion to something that gives an overview of the whole.

Comments on the Quality of English Language

The English language is easy to understand even for non-native speakers, and although it is written in simple sentences, it does not feel unrefined and I think it is relatively likable.

Author Response

Thank you for your kind comments 

We have edited the final line of the conclusion to de-emphasise the mention of drug eluting stents or other novel techniques. The remainder of the conclusion summarises the need for and the limitations in intraluminal therapy for UTUC as discussed in the review overall

Round 2

Reviewer 1 Report

Comments and Suggestions for Authors

The paper has been improved.

Comments on the Quality of English Language

seems to be good.